# LINKING FINITE-TIME LYAPUNOV EXPONENTS TO RNN GRADIENT SUBSPACES AND INPUT SENSITIVITY

## ABSTRACT

Recurrent Neural Networks (RNN) are ubiquitous computing systems for sequences and multivariate time series data. They can be viewed as non-autonomous dynamical systems which can be analyzed using dynamical systems tools, such as Lyapunov Exponents. In this work, we derive and analyze the components of RNNs' Finite Time Lyapunov Exponents (FTLE) which measure directions (vectors Q) and factors (scalars R) with which the distance between nearby trajectories expands or contracts over finite-time horizons. We derive an expression for RNN gradients in terms of $Q$ vectors and $R$ values and demonstrate a direct connection between these quantities and the loss. We find that the dominant directions of the gradient extracted by singular value decomposition become increasingly aligned with the dominant $Q$ vectors as training proceeds. Furthermore, we show that the task outcome of an RNN is maximally affected by input perturbations at moments where high state space expansion is taking place (as measured by FTLEs). Our results showcase deep links between computations, loss gradients, and dynamical systems stability theory for RNNs. This opens the way to design adaptive methods that take into account state-space dynamics to improve computations.

## 1 INTRODUCTION

Sequential inputs comes in a wide variety of forms, from natural speech or text, to electrical signals in the brain, financial markers, audio input and music, and more broadly, multivariate time series data (Pang et al., 2019; Mikolov et al., 2010; Das & Olurotimi, 1998; Tino et al., 2001; Su et al., 2020). Recurrent Neural Networks (RNN) specialize in processing such data by iteratively updating their hidden states $h_{t+1}$ based on previous states $h_t$ modulated by recurrent connectivity weights, and input $x_t$ via input weights. The compounding effect of signal amplification and dampening across many RNN iterations can lead to high sensitivity in some $h_t$ directions and very little in others, making training RNN over long sequential inputs challenging (Bengio et al., 1994). RNN form non-autonomous dynamical systems and thus, the evolution of the hidden states and memory of RNNs can be understood through the lens of dynamical systems theory and analysis. While RNN have been extensively studied, the relationship between state space dynamics — especially localized sensitivity to perturbations — and task performance remains relatively misunderstood. In this work, we explore this link and reveal new properties of RNN state space flows that inform, and can help guide, computations.

An important method for characterization of dynamical systems is *Lyapunov Exponents (LE)* (Ruelle, 1979; Oseledets, 2008) which measure the average separation/contraction rates of infinitesimally close trajectories. Recent work identifying RNN as dynamical systems has extended LE calculation and analysis to these systems (Engelken et al., 2020). Specifically of interest is the relation between Lyapunov exponent spectra and RNN performance as measured by network post-training accuracy, either by calculating the correlation between direct LE statistics and loss (Vogt et al., 2022a), or by training networks to capture a latent representation which separates networks according to accuracy (Vogt et al., 2022b). A limitation of this approach takes root in the fact that LEs are defined as asymptotic quantities, averaged as time approaches infinity and therefore, only capture averaged effects of space expansion and contraction. As a result, some more localized phenomena that may influence specific computations on particular input features are missed.

Here, we build on these previous works and derive connections between the gradient of the recurrent weight matrix and the finite-time intermediate values involved in LE computations over the course of a sequence. To do so, we leverage Finite Time Lyapunov Exponents (FTLE), quantities that have been initially derived to characterize fluid flows and the formation of Lagrangian Coherent Structures (Shadden et al., 2005; Haller, 2015; GREEN et al., 2007), time-dependent regions of state space that show high sensitivity to perturbations and act as dynamic separatrices. These methods are especially relevant in light of a recent resurgence in RNN-like methods to model long-term temporal structure called State Space Models, which have been demonstrated to exceed the capability of transformers while performing faster generation (Gu et al., 2022; Gupta et al., 2022a;b; Smith et al., 2023). Ongoing work translates these advances to new RNN structures which exhibit similarly impressive performance for very long sequences (Orvieto et al., 2023). Moreover, RNNs are often used to model neural circuits in neuroscience systems (Kaushik et al., 2022), for which measuring temporal and spatial patterns of activity simultaneously is necessary to fully understand modes of behavior (Barak, 2017). The lessons drawn from our work contribute to computational neuroscience by linking localized dynamic stability to task outcome.

Our contribution can be outlined as follows. We derive a novel expression of loss gradients in RNNs explicitly in terms of the components extracted in FTLE calculations, and we explore the evolution of both the FTLEs and their associated vectors to analyze their correlation and influence on performance and confidence on classification tasks. We demonstrate that the vectors associated with the maximal FTLEs of an RNN become further aligned over the course of an input sequence with the dominant modes of the gradient matrix as defined by the singular vectors. As a consequence of this, we show that the step-wise expansion and contraction of FTLE subspaces during Jacobian QR decomposition steps involved in LE computations can serve as indicators of input sensitivity. We show that selection of moments of high state space expansion are indicative of instabilities such that subtle input perturbations impact network output in significant adversarial ways. Through this work, we can develop a deeper understanding of information and gradient propagation in recurrent systems.

## 2 BACKGROUND AND MOTIVATION

### COMPUTATION OF LYAPUNOV EXPONENTS

Lyapunov Exponents can be computed by adopting the well-established algorithm (Benettin et al., 1980; Dieci & Van Vleck, 1995) and following the implementation of (Engelken et al., 2020), which principally relies on the QR matrix decomposition method. By definition, Lyapunov Exponents are an asymptotic quantity, but they can be estimated over very long iterates. A batch of input sequences $\mathbf{x}$ is sampled from a set of fixed-length sequences of the same distribution. For each input sequence in a batch, a matrix of $Q$ vectors, $\mathbf{Q}$, is initialized as the identity to represent an orthogonal set of nearby initial states. The hidden states $h_t$ are initialized as zeros.

The partial derivatives of the RNN hidden states at time $t$, $h_t$, with respect to the hidden states at time at $t-1$, $h_{t-1}$ form the Jacobian at time step $t$, $\mathbf{J}_t$.

$$[\mathbf{J}_t]_{ij} = \frac{\partial \mathbf{h}_t^j}{\partial \mathbf{h}_{t-1}^i}. \tag{1}$$

$\mathbf{J}_t$ is calculated and then multiplied by the vectors of $\mathbf{Q}_t$ to track the expansion and the contraction of the $Q$ vectors.

The *QR* decomposition of the Jacobian-Q matrix product is then used to retrieve an updated $\mathbf{Q}_{t+1}$ and expansion factors $\mathbf{R}_{t+1}$ at each time step:

$$\mathbf{Q}_{t+1}, \mathbf{R}_{t+1} = QR(\mathbf{J}_t \mathbf{Q}_t). \tag{2}$$

$r_t^k$ represents the expansion factor of the $k^{th}$ $Q$ vector at time step $t$ – corresponding to the $k^{th}$ diagonal element of $\mathbf{R}_t$ in the QR decomposition. Then, the $k^{th}$ Lyapunov Exponent $\lambda_k$ of the system with an input signal of length $T$ $(T \gg 1)$ is given by

$$\lambda_k = \frac{1}{T} \sum_{t=1}^{T} \log(r_t^k). \tag{3}$$

FINITE-TIME LYAPUNOV EXPONENTS

In order to adapt the above algorithm for FTLE, equation 3 is changed to:

$$\lambda_{k,t_f}^{FT} = \frac{1}{t_f} \sum_{t=1}^{t_f} \log(r_t^k) \tag{4}$$

Note that, in comparison to the Lyapunov Exponents (LE) in equation 3, the FTLE that we find in equation 4 are also a function of the intermediate time, $t_f$, as opposed to a fixed long sequence length $T$. This means that FTLEs are a sequence of values over time. Notably, this sequence is determined by the component parts of the QR decomposition used at each step in equation 2, $\mathbf{Q}_t$ and $\mathbf{R}_t$ (see Algorithm 1). We denote the columns of $\mathbf{Q}_t$ as the $Q$ *vectors*, where the $k^{th}$ column of $\mathbf{Q}_t$ is the vector associated with $\lambda_{k,t}^{FT}$. The FTLE calculated in equation 4 are ordered in dimension $k$, meaning that the first $Q$ vector at time $t$ corresponds to the largest FTLE, the second $Q$ vector corresponds to the second largest, and so on. We denote the diagonal elements of $\mathbf{R}_t$ as the *R values*, where $\mathbf{R}_t^{kk}$ represents the expansion or contraction factor in the $k^{th}$ dimension at time $t$, respectively.

---

**Algorithm 1:** Finite-Time Lyapunov Exponents Calculation

---

**for** $\mathbf{x}^j$ *in Batch* **do**
    initialize $\mathbf{h}$, $\mathbf{Q}_0$
    **for** $t = 1 \rightarrow T$ **do**
        $\mathbf{h} \leftarrow \mathrm{f}(\mathbf{h}, \mathbf{x}_t^j)$
        $\mathbf{J} \leftarrow \frac{d\mathbf{f}}{d\mathbf{h}}$
        $\mathbf{Q} \leftarrow \mathbf{J} \cdot Q_{t-1}$
        $\mathbf{Q}_t, \mathbf{R}_t \leftarrow qr(\mathbf{Q})$
        $\gamma_i \leftarrow \gamma_i + \log(R_{ii})$
        $\left[\lambda_t^{FT}\right]_i^j = \gamma_i^j / t$
    **end**
**end**

---

JACOBIAN RELATION TO GRADIENT

We present the problem of spectral constraints for robust gradient propagation, following the derivation introduced in (Vogt et al., 2022a). For transparent exposition, in this section we will consider the vanilla RNN, while the derivation is applicable to more complex RNN:

$$\mathbf{o}_t = \mathbf{W}\mathbf{h}_t, \;\; \mathbf{h}_t = \phi(\mathbf{a}_t), \;\; \mathbf{a}_t = \mathbf{V}\mathbf{h}_{t-1} + \mathbf{U}\mathbf{x}_t + \mathbf{b} \,, \tag{5}$$

where $\mathbf{V}$ is the recurrent weight matrix, $\mathbf{h}_t \in \mathbb{R}^N$ is the hidden state vector, $\mathbf{U}$ is the input weight matrix, $\mathbf{x}_t$ is the input into the network, $\mathbf{b}$ is a constant bias vector, $\phi$ is the non-linearity, and $\mathbf{W}$ is the output weight. The loss over $T$ iterates is the cumulative loss over each iterate $1 \leq t \leq T$. The loss at time $t$ is given by $L_t = f(\mathbf{y}_t, \hat{\mathbf{y}}_t)$, with $f$ some scalar loss function (e.g. Cross Entropy Loss), $\hat{\mathbf{y}}_t$ the prediction, and $\mathbf{y}_t$ is a target vector. The gradient of the loss in the space of recurrent weights $\mathbf{V}$, is given by

$$\nabla_{\mathbf{V}} L = \sum_{t=1}^{T} \sum_{i=1}^{N} \frac{\partial L}{\partial h_{t,i}} \nabla_{\mathbf{V}} h_{t,i} = \sum_{t=1}^{T} \mathrm{diag}(\phi'(\mathbf{a}_t)) \nabla_{\mathbf{h}_t} L \, \mathbf{h}_{t-1}^{\top} \,, \tag{6}$$

Here

$$\nabla_{\mathbf{h}_t} L = \sum_{s=t}^{T} \left( \prod_{r=t+1}^{s} \mathbf{J}_r^{\top} \right) \mathbf{W}^{\top} \nabla_{\mathbf{o}_s} L \,, \tag{7}$$

where $\nabla_{\mathbf{o}_s} L$ is an expression depending on the loss type (*e.g.* $\hat{\mathbf{y}} - \mathbf{y}_t$ for cross-entropy loss) and $\mathbf{J}_t = \frac{\partial \mathbf{h}_t}{\partial \mathbf{h}_{t-1}}$ is the Jacobian of the hidden state dynamics,

$$\mathbf{J}_t = \mathrm{diag}\left(\phi'(\mathbf{a}_t)\right) \mathbf{V} \,. \tag{8}$$

$\mathbf{J}_t$ varies in time with $\mathbf{x}_t$ and $\mathbf{h}_{t-1}$ via $\mathbf{a}_t$ and so is treated as a random matrix with ensemble properties arising from the specified input statistics and the emergent hidden state statistics.

## 3 GRADIENT MATRIX REPRESENTATION AND LINK TO FTLES

In previous sections, we described how FTLEs are computed. We present our derivation linking the components of FTLEs to RNN loss gradients, and ultimately to the loss.

### FTLE-GRADIENT RELATION

We derive the gradient of the RNN recurrent weight given by equation 7 in terms of the $Q$ vectors and $R$ values used in the calculation of FTLE. Given the algorithm for calculating FTLEs and $Q$-vectors requires multiplying the $Q$ vectors by the Jacobian $\mathbf{J}$ and then performing the QR decomposition, we can write the expression for the Q and R at time step $t$ in the following way:

$$\mathbf{Q}_t \mathbf{R}_t = \mathbf{J}_t^\top \mathbf{Q}_{t-1} \tag{9}$$

equation 9 can be used to solve for $\mathbf{J}_t$, allowing equation 7 to be recast by replacing the product of of Jacobians:

$$\prod_{r=t+1}^{s} \mathbf{J}_r^\top = \prod_{r=t+1}^{s} \mathbf{Q}_r \mathbf{R}_r \mathbf{Q}_{r-1}^{-1} \tag{10}$$

$$= \mathbf{Q}_s \left( \prod_{r=t+1}^{s} \mathbf{R}_r \right) \mathbf{Q}_t^\top , \tag{11}$$

using $Q^{-1} = Q^\top$ on the first $(r = t+1)$ $Q$ vectors with index $r - 1 = (t+1) - 1 = t$. When $t = 0$, the logarithm of the diagonals of the product of $\mathbf{R}$'s is equivalent to the FTLEs up to time $s$ before we divide by the time, $s$.

This gives the following expression for the full gradient with respect to the hidden state $h_t$:

$$\nabla_{\mathbf{h}_t} L = \sum_{s=t}^{T} \mathbf{Q}_s \left( \prod_{r=t+1}^{s} \mathbf{R}_r \right) \mathbf{Q}_t^\top \mathbf{W}^\top \nabla_{\mathbf{o}_s} L , \tag{12}$$

Notably, this novel derivation allows the use of the $Q$ vectors and $R$ values which emerge from the FTLE computation as a basis for the loss gradient of the hidden state (and by extension, the hidden weight matrix).

### LOSS GRADIENT MATRIX FOR V, $\Delta\mathbf{V}$

Let us consider the gradient of a loss function $L$, $\nabla L$. We are interested in comparing the updates to the hidden-state connection weights $\mathbf{V}$. We reshape the loss vector $\nabla_V L$ as a Tensor with the same shape as $\mathbf{V}$, such that the gradient vector has the shape $N \times H \times H$, where $N$ is the batch size, and $H$ is the hidden size. We will call this recast version $\Delta\mathbf{V}$. Finally, the subscript $t$ denotes the time step at which it was calculated.

After reshaping the gradient of the recurrent weights as $\Delta\mathbf{V}$, both $\Delta\mathbf{V}$ and the Jacobian $\mathbf{J}_t$ have the same shape. Over the course of training using Stochastic Gradient Descent with learning rate $\eta$, the $i^{th}$ update to the recurrent weight $\mathbf{V}$ takes the form $\mathbf{V}_{i+1} = \mathbf{V}_i - \eta\Delta\mathbf{V}_i$. In this form, the $\Delta\mathbf{V}$ operate in the same space as $\mathbf{J}_t$, and thus the $Q$ vectors (see Appendix A for more details).

## 4 EXPERIMENTS

In the previous section, we derived quantities that show the FTLEs and loss gradients are related in fundamental ways. We now set out to verify and exploit these links to show that state space dynamics have an impact on computations. Namely, we aim to show that the $Q$ vectors indicate the geometry of the gradient, aligning with the dominant modes of $\Delta\mathbf{V}$, and that the $R$ values represent the temporal contributions to the loss gradient, indicating the network's input sensitivity.

For our experiments, we consider a vanilla RNN (equation 5) trained on the sequential MNIST task. For such a task, the MNIST dataset of handwritten numbers is fed to the RNN the image as a sequence

of one or more pixels at a time, and the RNN must predict the number that was written at the end. In its standard form, the MNIST dataset images are $28 \times 28$ pixels, for a total of 784 pixels. We consider two different setups for this task.

1) Analysis of $\Delta \mathbf{V}$ and the geometric alignment with $Q$ vectors: We compute the cosine similarity (alignment) between singular vectors of $\Delta \mathbf{V}$ and the $Q$ vectors and demonstrate the evolution of this alignment over the course of training and sequence index as the confidence of the network's predictions increases. We consider a smaller network (128 hidden units) trained on shorter sequences, given the significant computational cost of calculating the gradient at each step. For this experiment, we consider the row-wise SMNIST task (Deng, 2012), in which the network receives a full row of the image (28 pixels). Since there are 28 rows in each image, the sequence length for this setup is 28.

2) Analysis of $R$ values and input sensitivity: We identify input locations to perturb based on times at which the greatest degree of expansion in state space occurs. We demonstrate the impact these perturbations have on network output is greater than that of perturbations at randomly-selected input locations. We use a larger RNN (512 units) trained on MNIST dataset with permuted pixel-wise input, giving an input length of 784. This size of network is necessary to achieve good performance on sequences of this length. Furthermore, by reducing the dimension of the input, we are able to isolate the individual input pixels which correspond to the $R$ values at that point in the sequence.

### ALIGNMENT OF GRADIENT WITH $Q$ VECTORS

When computing the singular value decomposition (SVD) of $\Delta \mathbf{V}$, the first singular vectors capture the dominant modes of the gradient. We assume these first vectors correspond to the directions in parameter space which correspond to the greatest loss increase, with later vectors capturing less of this effect. Meanwhile, the $Q$ vectors are also ordered at each time step to reflect the direction of decreasing expansion/increasing contraction as the index increases. In effect, for high-index $Q$-vectors, the rate of contraction is higher, meaning that information stored along that direction is rapidly forgotten, whereas earlier indices have a greater degree of information preservation, and even extra sensitivity if expansion is pronounced.

To test the degree of alignment between the $Q$ vectors ($q_t$) and the singular vectors of $\Delta \mathbf{V}$ ($e_{V_t}$), we take the inner product of all pairs of vectors between each set. As both of these sets of vectors are respectively orthonormal, this inner product yields the cosine similarity, a value between -1 and 1 indicating the degree to which the vectors are parallel/aligned. 1 indicates parallel, 0 indicates orthogonal, and $-1$ indicates antiparallel. We store these alignment values in a matrix $A_t$.

$$[A_t]_{ij} = q_{ti}^\top \left( e_{V_t} \right)_j \tag{13}$$

Since the number of hidden states of the RNN under consideration is 128, the $Q$ vectors and gradient singular vectors are each 128-dimensional. In this high-dimensional space, the probability of two randomly-selected variables being effectively orthogonal is very high, due to concentration of measure on the sphere (Talagrand, 1996). Thus, most vectors in this high-dimensional space are expected to have alignment values close to 0.

The $Q$ vectors are calculated for twenty five different initializations of $h_0$ sampled from a standard normal distribution, along with the loss and gradient associated with this initial hidden state. Eq equation 13 is then used to find the alignment across all combinations of $Q$-vectors and gradient singular values.

We begin by analyzing the dominant directions of $\Delta \mathbf{V}$ measured by the first five singular vectors. To study the level of alignment with the $Q$-vectors, we show histograms of the alignment values between these first five singular vectors and the first ten, the last ten, and ten randomly-selected $Q$-vectors across all time steps in Figure 1.

Whereas the distributions of alignment are similar across these three sets for an untrained network, once the network has been trained for 5 epochs, the distributions of alignment between the dominant directions of $\Delta \mathbf{V}$ and each set of $Q$ vectors differ. The first ten $Q$ vectors, corresponding to the ten largest FTLEs, have a wider distribution, indicating a greater number of vectors aligned with the first five singular vectors of $\Delta \mathbf{V}$. As a result, the standard deviation of this distribution increased from 0.089 when untrained to 0.200 after 5 epochs. In contrast, the distribution of the alignment between

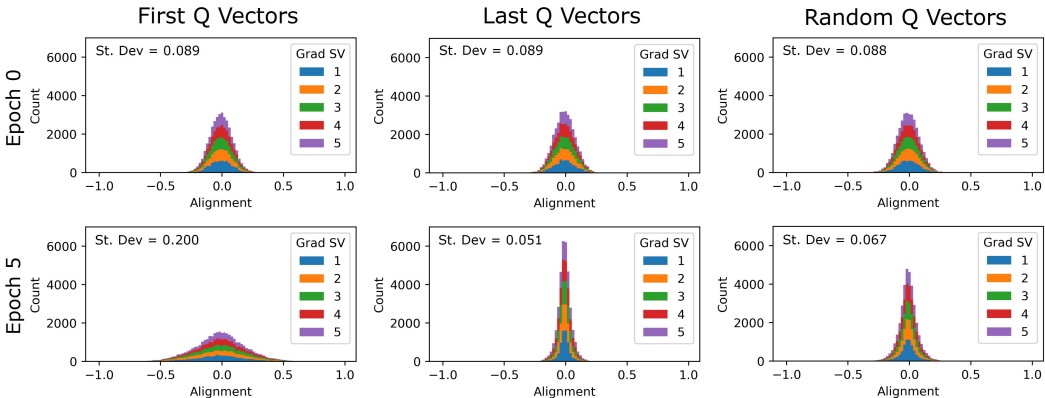

Figure 1: Distribution of alignment values (as stacked histogram) at epoch 0 (top row) and epoch 5 (bottom row) across all time steps between the first 5 Singular Values of $\Delta \mathbf{V}$ and three sets of ten $Q$ vectors: The first ten (left column), the last ten (middle column), and ten randomly selected indices not including the first or last ten (right column). The standard deviation of each cumulative distribution is indicated in the top-left of each plot.

the last ten $Q$ vectors and first five singular vectors of $\Delta \mathbf{V}$ becomes narrower, concentrating more around 0, with its standard deviation decreasing from $0.089$ to $0.051$. The set of randomly-selected $Q$ vectors has a slightly narrower and taller peak around zero than it did at epoch 0, causing the standard deviation to decrease from $0.088$ to $0.067$.

To study how the alignment of these vectors changes depending on the $Q$-vector index and the step in the sequence, we show in Figure 2 the standard deviation of the distribution of alignment values for the first five and last five singular vectors of the gradient $\Delta \mathbf{V}$ for each $Q$ vector index over a sequence.

As the network trains, a clear structure emerges in the standard deviations of the alignment values as a function of the $Q$ vector index and sequence time step. As seen in Figure 2, before training (Epoch 0), the standard deviation of the alignment with the first and last singular vectors of the gradient at each $Q$-vector index fluctuates around a mean value consistent with the standard deviation for the randomly-selected $Q$ vectors (see red line). This is true both early in the input sequence (purple dots) and towards the end of the sequence (yellow dots). However, once the network has been trained, the standard deviation of the alignment between the first five singular vectors of $\Delta \mathbf{V}$ and the first several $Q$-vectors is much greater than the average value for the random indices. Moreover, there is a sharp decrease in the standard deviation of the alignment as the $Q$-vector index increases. Both of these effects become more pronounced later in the sequence, leading to a gradual decreasing curve over $Q$ vector index at the final step in the sequence. Notably, for $Q$ vectors after index 40, the standard deviation at the end of the sequence decreases to well below that of the random vectors, indicating that these directions are increasingly orthogonal to the dominant directions of the gradient.

Meanwhile, the standard deviation of the alignment with last five singular values of the gradient has a similar but mirrored pattern. Once the network has been trained, the first several $Q$-vectors are less aligned and therefore more orthogonal to these singular vectors. Then, the alignment gradually increases for approximately the first twenty $Q$ vectors until reaching the same average baseline that the untrained network had.

Through this analysis, we find that the basis of $Q$ vectors reveal the directions in hidden space which are more aligned with and which are more orthogonal to the dominant modes of the gradient update. Additionally, we find that the least informative modes of the gradient have effectively random alignment with all $Q$ vectors except the first few, with which it is more orthogonal. This shows that state-space dynamics and its sensitivity to inputs is shaped throughout training in a manner that aligns with directions that are relevant to the task, as measured by the loss' gradient. We now investigate how state-space expands or contracts along these directions, and how these transformations are related to computations.

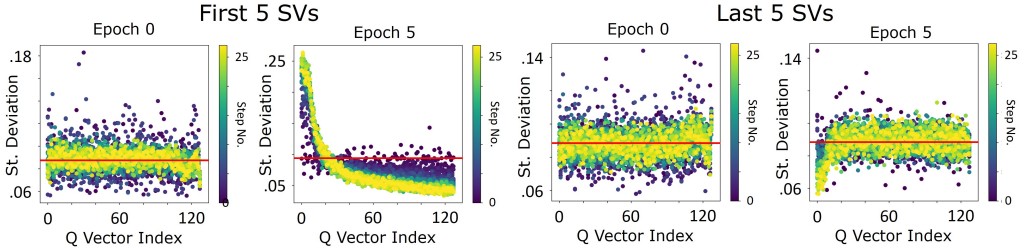

Figure 2: Standard deviation of alignment values as a function of $Q$-vector index and sequence step number before and after training. Alignment is calculated between each $Q$-vector and the first five singular vectors of $\Delta \mathbf{V}$ (left) as well as the last five singular values of $\Delta \mathbf{V}$ (right) for each time step in the sequence. The average standard deviation of the alignment values for randomly selected vectors is shown as a red horizontal line.

SENSITIVITY ANALYSIS OF $R$ VALUES AND INPUT

For this experiment, we consider the pixel-wise SMNIST task, since we want to analyze the sensitivity to individual inputs. The input size is 1 pixel, and the input sequence length is 784, with some fixed permutation of the pixels. We train an RNN with a hidden size of 512 units and initialize the weight matrices with the Xavier normal initialization. The loss used for this task is cross entropy loss. For the computation of FTLEs to extract the $R$ values, we use the default PyTorch initialization for the initial hidden states of the RNN, which is setting $h_0$ to all zeros.

We study the expansion and contraction factors, $R$ values ($r_t^k$ in equation 4), throughout the sequence, and demonstrate that they can indicate the network's sensitivity to input. We perform this analysis by comparing the predictions of the network based on the original input, versus input which has been perturbed based on a threshold which depends on the ordering of $R$ values.

We observe that ordering pixels according to the value of the first $R$ value at the time where the pixel was presented ($r_t^1$) has the most negative slope, indicating a greater distinction between successive indices (see supplemental materials for more details).

Given we know these $R$ values indicate the running contribution to exponential expansion/contraction up to time $t$, we hypothesize that the first $R$ value, $r_t^1$ is a good indicator to predict which input pixels are most sensitive to perturbations and thus, contain more discriminatory power. Going forward, we denote $R_1 = r_t^1$.

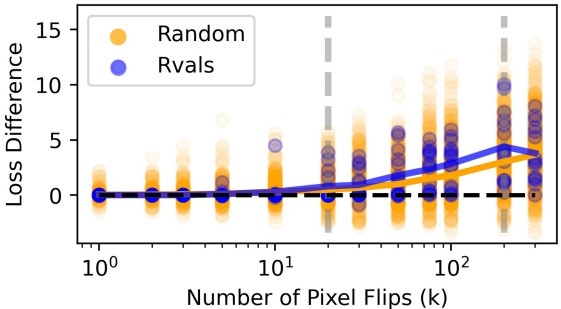

Figure 3: Difference in Loss Values between initial input and flipped inputs as a function of the number of "up" (blue) and "down" (orange) flips. When flipping between 20 and 200 pixels (see vertical dashed lines), the increase in loss is greater when the flips are based on $R$ values rather than randomly-selected (see Table 1).

Using this ordering of input indices, we select a number of pixels we wish to perturb to test the sensitivity of the network. The pixel corresponding to the index of the $k$ largest $R_1$ values over a sequence are then chosen to be "flipped". If a candidate pixel is chosen to "flip", we do the following: If the grayscale value of the pixel is non-zero, we set the grayscale value to zero (we call this a "down

flip"); If the grayscale value of the pixel is already zero, we instead increase it to the maximum value of 1 (we call this an "up flip").

For comparison, we also randomly select samples of $k$ pixels from the input to flip. For each choice of $k$, we select 50 samples of $k$ random pixels on which to perform flips. To determine the relative sensitivity of the network to these two choices of perturbation, we calculate the difference in the network loss between the original input and the perturbed inputs (with $k$ flipped pixels). This is performed over 15 different original input images.

The difference in network loss (defined as the network's cross entropy loss on the perturbed input minus the loss on the original input) as a function of the number of pixels flipped, $k$, is shown for each sample as an individual dot in Figure 3. Additionally, the mean loss difference across all samples for a given $k$ is shown as a line of the same color.

Table 1: Statistics of loss differences between $R$ value-flipped and randomly-flipped inputs

| k | $\mu_R - \mu_{\text{rand}}$ | Variance | p-value |
|---|---|---|---|
| 1 | -0.028 | 0.009 | 0.998 |
| 5 | -0.008 | 0.084 | 0.540 |
| 10 | 0.025 | 0.303 | 0.468 |
| 20 | 0.313 | 0.341 | 0.179 |
| 50 | 0.794 | 0.523 | 0.065 |
| 75 | 0.814 | 0.475 | 0.043 |
| 100 | 1.050 | 0.530 | **0.024** |
| 200 | 1.363 | 0.824 | 0.049 |

For $20 < k < 300$, the mean loss difference is greater when flipping according to $R$ values as opposed to random pixels. We investigate the significance of the difference between these losses by performing a p-test on the hypothesis for each $k$ that the mean of the $R$ value loss ($\mu_R$) is greater than mean of the random-selection loss ($\mu_{\text{rand}}$). We show the results of this test ($\mu_R - \mu_{\text{rand}} > 0$) in Table 1. The smallest p value is achieved for $k = 100$, at $p = 0.024$, but we find that $k = 75, 100$, and 200 pixels, we get $p < 0.05$.

To illustrate this sensitivity, we show the result of the network predictions as a result of selecting 100 pixels to flip corresponding to the locations where $r_t^1$ is the largest. We show in Figure 4 the input images after these 100 flips are performed according to the largest $r_t^1$. The "up flips" lead to more bright (yellow) pixels which used to be dark (purple), while the "down flips" lead to more dark pixels which used to be lighter (blue, green, or yellow). Furthermore, we show how the network predictions resulting from the original input (black), the perturbed input according to $R$ values, and the randomly-perturbed inputs. As expected, the flipping according to the $R$ values leads to changed predictions by the network, and to a greater extent than the average across the randomly-perturbed inputs. We demonstrate in two examples that, even when the network originally predicts the correct label and consequently has very low loss, flips on these 100 selected pixels can cause the network

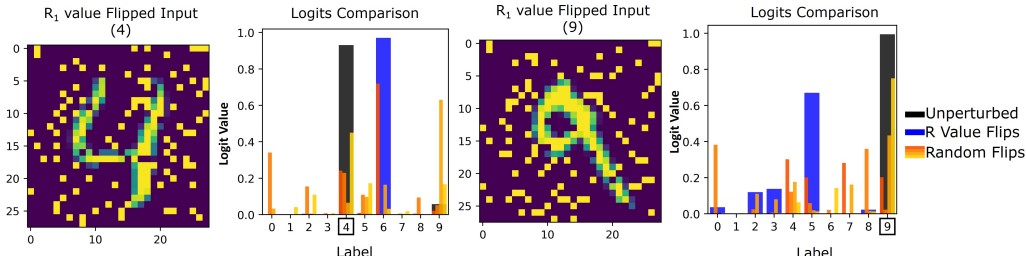

Figure 4: Comparison of network output logits between original input and perturbed outputs for two sample images. The perturbed input has had $k = 100$ pixels flipped from the original input image. The bar plots over 10 digit classes show network output logits for each example input for different perturbation scenarios: unperturbed input (black), the $R$ value-flipped input as shown (blue), 4 examples of 100 randomly-flipped pixels (red, red-orange, orange, yellow).

to predict a new (and incorrect) label with high confidence. Furthermore, we find the resulting confidence in this incorrect label is much greater than for random perturbations, which tend to have lower confidence for incorrect logits.

## DISCUSSION

In this work, we demonstrated that the stability of state space dynamics in RNNs is functionally linked to task computations and can help identify features in dynamics that are crucial for performance. Indeed, we show that increments in expansion/contraction rates that are used to estimate FTLEs, together with the orthogonal directions associated with the linearization of dynamics, are representative of directions in neural activity space for which loss gradients are highly sensitive. Our arguments rest on the fact that the $Q$ vectors at some time $t$ indicate the directions associated with the expansion or contraction factors given by the $R$ values. In analytic derivations, we show that the gradient of the hidden state can be expressed explicitly using a basis of the $Q$ vectors and $R$ values, as shown in equation 12.

Through complementary numerical experiments, we validated that the $Q$ vectors can capture hierarchically the geometry of the gradient. $Q$ vectors associated with the greatest degree of expansion aligned with the dominant directions of the gradient, as measured by the singular values of $\Delta \mathbf{V}$. We further validated that rate of expansion and contraction in the state space measured by the $R$ values leads to increased input sensitivity. This manifests in the form of perturbation effects timed at moments of heightened state space expansion. Indeed, when perturbed at moments corresponding peaks in aggregate effects of the $100$ largest $R$ values, the impact on the network's loss was significantly more pronounced than for random perturbations. Beyond loss value, such perturbations lead to reliably wrong digit classification in sequential MNIST tasks with very high confidence and measured by logit magnitude, a phenomenon unobserved when randomly perturbing.

In sum, the geometric picture provided by the $Q$ vectors characterizes gradient propagation, and temporally localized $R$ values link state space expansion to task uncertainty. Since the first $Q$ vectors become increasingly aligned with the dominant gradient modes as the network confidence increases (both over training epoch and over an input sequence), it would seem that greater expansion in state space correspond to more definite network outputs. This means that changes that impact these moments have the greatest impact on the loss, showing that sensitivity in dynamics translates to a bigger impact on credit assignment. Thus, these directions can be interpreted as "ridges" along which the network's sensitivity is greatest. Such an interpretation is analogous to the ridges found in the study of Lagrangian Coherent Structures (Shadden et al., 2005), but of the gradient as opposed to the state space.

Further investigations into how FTLE and related quantities can be leveraged to improve and/or analyze RNN training are warranted. The decomposition of the gradient into these components parts seems to be a promising direction for further development, and could lead to novel regularization strategies. Furthermore, the method presented for analyzing $R$ values to detect points of momentary sensitivity could be leveraged to design localized (in time) credit assignment. For example, instead of computing gradients from entire backprogated-through-time trajectories, one could foresee an adaptive method which focuses on parameter updates that alleviate large expansion/contraction episodes, which we show correlate with performance.

In addition, beyond the ML/AI applications where RNNs have given way to transformers in many applications, RNNs are extremely relevant to the field of Neuro-AI where the analysis of neural circuits (which are recurrent) tries to uncover ways by which brains perform computations. Our results contribute to a long line of dynamical analyses for neuroscience-relevant models (Lajoie et al., 2013; Farrell et al., 2022; Sussillo, 2014).

While outside the scope of this work, regularization of the gradient to encourage greater alignment with the first few $Q$ vectors, analysis of the per-neuron contributions to the gradient directions through the $Q$ vectors, introducing an attention mechanism that depends on the $R$ values, or generating adversarial learning examples based on the largest $R$ values, could be interesting extensions.

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
