# OpenReview forum: "Linking Finite-Time Lyapunov Exponents to RNN Gradient Subspaces and Input Sensitivity"
_ICLR.cc/2024/Conference — Submitted to ICLR 2024_

### Official Review · Reviewer_YMuh · 2023-10-20

**Soundness:** 2 fair
**Presentation:** 3 good
**Contribution:** 3 good
**Rating:** 5
**Confidence:** 3

**Summary:**

Considering RNNs, the manuscript shows how the directions of strongest expansion in state space, as quantified by the finite-time Lyapunov exponents, become during training related to the larges singular values of the gradient matrix (gradient vector in weight space reshaped into NxN matrix in activation space). The authors explore the consequences of this link, arguing that time-points followed by strong (finite-time) expansion and the associated directions of maximal expansion structure the sensitivity of the network to perturbations.

**Strengths:**

- Elucidating how chaotic dynamics shapes computation in recurrent networks is a relevant and challenging topic, on which the manuscript provides a novel perspective.
- Novel observation that the gradient structure and (FT)Lyapunov vectors become aligned during training.
- Mostly clear presentation.

**Weaknesses:**

- Fig 3 + Table 1 are not convincing.
- While the manuscript shows how the gradient can be written in terms of the FTLE and associated directions, the following observations (alignment during training, flip of prediction with high logit confidence after expansive perturbations) are numerical and not directly derived from the mathematical part. As a consequence, where the numerical evidence is not entirely convincing (Fig 3, 4), the conclusions seem to be resting on rather weak support.

**Questions:**

- Why are the hidden states initialized to h_t = 0 in the section analyzing sensitivity, but to nonzero in Fig1+2 ?
- Below eq.8 : states that J_t is treated as a random matrix, but later the analysis is based on specific instances of (trained) weights ?
- Why not put the FTLEs explicitly into the expression in eq.12 ?
- Why is the cost of computing the gradients in each step considerable (p.5)? Autodiff libs do this fast after all. Or is the QR+SVD decomposition the expensive part?
- Given eq.12, is there a mechanistic way to see that an increasing alignment of Delta_V and Q should appear over training time?
- In Fig.1 a bit unclear: distribution of alignments between first 5 vectors of Delta_V and given subset of Q vecs, across 25 different initializations of h_0: Does this mean that the initialization of V is fixed, but then 25 models are trained with different h_0 (same for all data samples)? Or is h_0 varied in the already trained system?
- Alignment is created after 5 epochs as shown in Figs 1+2, does it vanish again when trained to saturation? (One could expect that the gradients get smaller and more noise dominated?
- p.7: "[...] which pixels are most sensitive to perturbations and thus, contain more discriminatory power" The term discriminatory power seems misplaced here, since the network output is expected to be sensitive to these pixels, but these pixels are not hypothesized to contain more information about the label?
- Fig 3 / Table 1 where not convincing to me due to the very small difference and large variance; unclear if the effect is robust. The p-values should be considered with care due to (i) the multiple testing problem and (ii) post-hoc selection of the tested k-values. Can this difference of the curves be measured beyond doubt with small standard deviations (delta > 3 sigma)? Furthermore, why is there no difference between the two curves for smaller number of flips?
- For a large network, would it be expected that the largest R-value becomes self-averaging, and there are little differences across time-steps/pixels?
- Fig 3: blue/orange means up/down or R-value/random based flips?
- Fig 4: The difference in logit size between R-value flip and random flip seems interesting, but is not statistically quantified. Is there a theoretical argument why this behavior could be expected?
- In the discussion, an increased alignment of first gradient singular values and first Q vectors is claimed to appear both over training epoch and over input sequence, but figures or references in the main text seem to show this only for training epoch 5, not for later epochs and not for later positions in the input sequence.

---

### Official Review · Reviewer_kefG · 2023-10-30

**Soundness:** 3 good
**Presentation:** 3 good
**Contribution:** 3 good
**Rating:** 5
**Confidence:** 3

**Summary:**

The paper studies how, during the training of an RNN, the activation space trajectories of the RNN evolve. Specifically, the authors propose Finite Time Lyapunov Exponents (FTLE) to measure how the direction and magnitude of nearby activation space trajectories expand. Finally, the paper tries to build a connection between the activation space dynamics of the forward phase and the loss/gradient dynamics in the backward phase of the training.

**Strengths:**

- A novel idea in an important topic of an underexplored area (linking forward and backward dynamics especially)
- Theoretically and experimentally sound paper

**Weaknesses:**

- The main limitation I see with this work might be its real-world relevance. While the approach shows interesting relations in the training process, the question this opens is how to build RNN architectures, training algorithms, regularization terms, loss functions etc. that can improve the training of RNNs
- Related to the point above, it would be interesting to see how observed dynamics/relations differ or are consistent across different RNN architectures, loss functions, etc.

**Questions:**

What are some of the implications of the observed dynamics for building better RNNs or training algorithms?

---

### Official Review · Reviewer_NbRp · 2023-11-01

**Soundness:** 2 fair
**Presentation:** 2 fair
**Contribution:** 2 fair
**Rating:** 3
**Confidence:** 3

**Summary:**

This paper studies the connection between finite time Lyapunov exponents (FTLE) calculation and RNN loss gradients. More specifically, they derive the expression of loss gradients of RNN using terms ($\mathbf{Q}$ and $\mathbf{R}$) from FTLE calculation. Empirically, they conducted analysis using MNIST dataset, where they demonstrate that the geometry of gradients of loss are aligned with $Q$ vectors. They further conducted sensitivity analysis using the $R$ values.

**Strengths:**

The overall presentation is clear and easy to follow.

The focus on the connection between Lyapunov exponents and training dynamics of deep neural networks are important and interesting.

**Weaknesses:**

W1. The novelty of the paper is constrained in the lack of reviews of related works. The connection of RNN to the Lyapunov exponents are well studied in [1]https://arxiv.org/abs/2110.07238; [2]https://arxiv.org/abs/2306.04406. A detailed comparison is necessary to evaluate the novelty of the paper's claim.

W2. Empirically, the authors should consider experiments on dynamical data instead of using only MNIST.

W3. From the result presented in Figure 3, it is not sure how useful the sensitivity analysis using R values would be: we cannot get rid of retraining when you vary the number of pixel flips.

W4. Notations are not consistent. For example, in part 2, the un-bolded symbol of $h_t$ and the bolded one $\mathbf{h}_t$ are used in the same time to denote the gradients vector.

**Questions:**

Please see the main concerns listed in W1, W2 and W3.

---

### Official Review · Reviewer_pkxF · 2023-11-06

**Soundness:** 2 fair
**Presentation:** 3 good
**Contribution:** 2 fair
**Rating:** 5
**Confidence:** 3

**Summary:**

In this paper, empirical analyses are performed that relate the lyapunov vectors of the hidden state dynamics of an RNN to the gradient updates. The authors observed that the finite-time backward Lyapunov vectors are more aligned with the top singular vectors of the gradient updates in trained networks. In a related observation, they see that perturbations to the inputs at times of largest local-in-time state-space expansions (as determined by R values) have a larger effect on the loss function than random perturbations.

**Strengths:**

Thorough experiments and detailed explanation of experimental setup.

**Weaknesses:**

While the empirical results are interesting, they warrant a deeper discussion that makes use of well-understood theory of Lyapunov vectors.
By employing existing theoretical results (see the question section), it may be that the numerical observations are not surprising.
The authors should look into adjoint Lyapunov vectors -- orthonormal bases for the dual or the adjoint of the tangent spaces. These spaces essentially characterize the growth/decay of sensitivities/derivatives of an observable of the dynamics with respect to state space perturbations. Using one definition of adjoint vectors, we can check that an adjoint basis vector of an index $i$ is orthogonal to backward Lyapunov vector of index $i \neq j.$

The FTLEs obtained with an adjoint basis (fixing some observable, which here is the loss function) are the same as those obtained with tangent space algorithms (such as the QR iteration carried out in the paper). This may explain all the experimental results of alignment and also the observations of maximum sensitivity of the loss to perturbations applied with $R$ values are large.


Please note that the original reference for QR-based Lyapunov vector calculations is Ginelli et al 2007 PRL.

**Questions:**

Main question: Can one apply the knowledge of the dual basis lyapunov vectors (fixing the loss function as an observable of the hidden state dynamics) and their properties (see e.g. Ginelli et al 2007, Ginelli et al 2013, Kuptsov and Parlitz 2012 and several references for the theory of Lyapunov vectors) to explain the experimental observations?

---

### Meta-Review · Area_Chair_BQ41 · 2023-12-05

**Metareview:**

This paper explores the dynamics of Recurrent Neural Networks (RNNs) during training using Finite Time Lyapunov Exponents (FTLE). The reviewers acknowledged the potential interest in studying training dynamics and the connection between forward and backward dynamics. However, they also identified various concerns. Regrettably, the authors did not engage in the rebuttal phase, forfeiting the chance to address these concerns. Considering the authors' lack of response and the overall critical assessment by the reviewers, I recommend rejection of this submission.

**Justification For Why Not Higher Score:**

Authors didn't engage in the rebuttal phase to address concerns raised by the reviewers. I agree with the overall assessment of the reviewers.

**Justification For Why Not Lower Score:**

N/A

---

### Decision · Program_Chairs · 2024-01-16

Reject